# The Influence of Packing Methods and Storage Time of Poultry Sausages with Liquid and Microencapsulated Fish Oil Additives on Their Physicochemical, Microbial and Sensory Properties

**DOI:** 10.3390/s21082653

**Published:** 2021-04-09

**Authors:** Krzysztof Kawecki, Jerzy Stangierski, Renata Cegielska-Radziejewska

**Affiliations:** Department of Food Quality and Safety Management, Faculty of Food Science and Nutrition, University of Life Sciences, Wojska Polskiego 31, 60-624 Poznan, Poland; renata.cegielska-radziejewska@up.poznan.pl

**Keywords:** chicken sausages, fish oil, microencapsulated fish oil, vacuum packing, modified atmosphere packing, physicochemical properties, microbiological analysis, sensory assessment

## Abstract

The aim of the study was to evaluate the influence of refined fish oil additives in liquid and microencapsulated forms, packing method (VP—vacuum packing, MAP—modified atmosphere packing) and storage time (1, 7, 14, 21 days) on selected physicochemical, microbiological and sensory characteristics of minced poultry sausage. Principle component analysis (PCA) showed that the fish oil additive, packing method and storage time significantly influenced some of the physicochemical characteristics of the sausages. The pH value was negatively correlated with the type of sample and packing method. The water activity decreased along with the storage time. The sausages with microcapsules had distinguishable hardness, gumminess and chewiness than the other samples. This tendency increased in the subsequent storage periods. The packing method and storage time of the samples had a statistically significant influence on the growth of the total colony count and count of lactic acid bacteria (*p* < 0.05). The most aerobic bacteria were found in the control sample, and the least in the sample with microcapsules, regardless of the packing method. The use of MAP and the addition of microcapsules resulted in the lowest microbiological contamination of the sausages. The sensory analysis made by a trained panel did not show any significant differences between the samples. After 21-day storage of the sausages there was a slight decrease in some of the sensory parameters, e.g., color, smell, taste. The liquid oil and microencapsulated oil additives in the meat filling did not negatively affect the taste or any physicochemical characteristics of the meat products. From the microbiological perspective, there were better effects from the MAP method.

## 1. Introduction

Meat consumption is affected by the standard of living, diet, livestock production, and consumer prices, as well as macroeconomic uncertainty and GDP shocks (Gross Domestic Product). In 2019, the average global consumption of poultry meat was 14.75 kg per capita. According to forecasts for 2020, a further but much slower increase in global consumption was predicted, i.e., only 0.1% [1]. In 2020 the consumption of poultry meat in the EU was predicted to increase to 1.5%, i.e., 23.7 kg per capita [2].

Poultry meat is one of major sources of protein, with a well-balanced amino acid composition. Poultry meat lipids are another essential ingredient and these are primarily a mixture of triacylglycerols, cholesterol and phospholipids. The nutritional value of poultry meat products can be further improved by increasing the content of polyene fatty acids, which will result in a favorable ratio of fatty acids, especially those of the n-3 and n-6 families. This effect can be achieved through modifying the composition of meat products by replacing animal fat with vegetable or fish oils [3,4,5]. There have been studies where animal fat was replaced with liquid fish or vegetable oil [6] stabilized in a konjac matrix, or with soy protein and pre-emulsified caseinates [7], or with milk proteins and carrageenan [8]. There have also been studies in which the fat composition in lunch meats was modified with microcapsules filled with appropriate n-3 polyunsaturated fatty acids (PUFA) [4,9]. These strategies boil down to increasing the content of beneficial compounds and limiting those with negative health effects in meat and meat products. These approaches may significantly affect animal production practices (genetic and nutritional) and meat transformation systems [10]. The use of healthier ingredients and modified fat profiles (lower content of saturated fats, cholesterol, and higher content of polyunsaturated fats) remain issues to be investigated, and the meat industry must adapt to these requirements [11]. Reduced fat content and modified fat profiles positively influence purchase decisions and health perception, and surprisingly, these reformulations do not have a significant negative effect on taste expectations [12]. 

It is important to use the appropriate packaging to preserve the quality and safety of modified finished products. The food packing function has evolved from simple passive preservation methods to active and intelligent ones, which go beyond traditional functions [13]. Vacuum packing (VP) and low-oxygen modified atmosphere packing (MAP) are widely used in the meat industry. Vacuum packing, which eliminates air without replacing it with another gas, prevents product contamination and loss of water. The MAP method consists of replacing air with a mixture of gases. Different proportions of N_2_, CO_2_, and O_2_ are used for lunch meat products [13,14,15]. These methods of cold meat packing extend the shelf life, increase availability during distribution and improve the presentation of processed products on refrigerator shelves [16].

The undesirable growth of bacteria in meat products depends on the storage conditions, i.e., temperature, and the type of package [17]. The final quality of stored lunch meat is also influenced by the degree of enzymatic degradation [18] and the oxidation of lipids and proteins, which decrease the color stability, flavor, and tenderness [19]. Lactic acid bacteria (LAB) are thought to be one of the major causes of spoilage of heated and vacuum-packed meat products. The growth of these bacteria causes specific types of spoilage of certain foods, such as a sour smell, lower pH, mucus production, and other spoilage that shortens the shelf life of meat products [20]. For this reason, it is important to assess the behavior of LAB and the total count of microorganisms during the storage of meat products.

Industrially produced poultry sausage types such as the frankfurter, which were tested in our study, have a favorable composition (protein—about 18%, fat—about 11%) and are popular and highly rated by consumers in Poland and other countries. Currently, some lunch meat producers are attempting to meet the recommendations of the World Health Organisation [21] concerning lipid intake and are offering products with a lower content of saturated fats and a higher amount of unsaturated fats. This study follows these trends. The research was conducted on vacuum-packed (VP) and modified-atmosphere-packed (MAP) finely comminuted poultry sausages with liquid and microencapsulated fish oil additives, which were stored in a refrigerator for 21 days. It was important to obtain modified sausages whose sensory characteristics would not be worse than those of the standard products. Selected physicochemical, microbiological, and sensory characteristics of the products were assessed to determine how they changed in particular periods of refrigerated storage.

## 2. Materials and Methods

The aim of the lunch meat producer was to obtain poultry sausages containing an adequate level of n-3 acids to declare a product that was ‘High in omega-3 fatty acids’, in accordance with the Commission Regulation (EU) No 116/2010 of 9 February 2010 of the European Parliament and of the Council with regard to the list of nutrition claims. The minimum required amount was at least 80 mg of eicosapentaenoic acid (EPA) and docosahexaenoic acid (DHA) in total per 100 g and 100 kcal of the product.

### 2.1. Sample Preparation

Model sausages were made from small pieces of chicken breast (60%), boneless and skinless chicken legs (18%) and chicken skins (22%). A typical set of functional additives was used: salt (1.4%), sodium ascorbate (0.1%), spices and aromas (3.2%), and water (20%). The raw meat was comminuted in a grinder with a mesh diameter of 5 mm. Everything was mixed thoroughly in order to distribute all the ingredients evenly. The resulting stuffing was divided into three parts: a control sample (CO), a sample with an oil additive (FO), and a sample with a microencapsulated oil additive (MC). The last ingredient added to each 30-kg portion of the stuffing was an appropriate amount of liquid fish oil and microcapsules with n-3 acids. The final temperature of the stuffing was 8–10 °C.

Two oil preparations made by ^©^DSM Nutritional Products Ltd., Basel, Switzerland, were used in the study. These were: MEG-3^TM^ 30% 8a Food Oil consisting of refined fish oils, and fish oil microencapsulated in pork gelatin MEG-3^TM^ 30% Powder. According to the manufacturer’s certificate, the total content of EPA and DHA in the MEG-3™ 30% 8a food oil preparation was 25%. The total content of EPA and DHA in the MEG-3™ 30% powder preparation was 15%.

The data listed above, raw material composition, and the manufacturer’s declaration of the EPA + DHA content in the oil preparations were used to set the amount of MEG-3™ 30% 8a Food Oil and MEG-3™ 30% Powder at 7.1 g kg^−1^ and 11.9 g kg^−1^ of the batter, respectively. The mean efficiency of the sausage production process was about 86%. The inclusion of EPA and DHA in the oil and powder formulation resulted in 2.0 g kg^−1^ of fatty acids in the finished products.

The resulting three variants (CO, FO, MC) were vacuum-packed (VP) and modified atmosphere packed (MAP) with 5 sausages in each package. Next, they were stored in a refrigerator at a temperature of 4 °C ± 2 °C for 21 days. Initially, the gas mixture was composed of: 0.06% O_2_; 72.0% N_2_, and 27.9% CO_2_. All the parameters (except for the basic composition and sensory evaluation) were measured on days 1, 7, 14, and 21 of refrigerated storage.

### 2.2. Basic Chemical Composition

The following methods were applied to determine the basic chemical composition of the samples: dry matter—the oven method [22]; total protein content—the Kjeldahl method, where 6.25 was the conversion factor value [23]; total fat content—the Soxhlet method [24]; ash content [25]. The composition was analyzed during the first period of refrigerated storage for the sausages.

### 2.3. pH Measurements 

The pH of the samples was measured with a digital portable HI 99,161 pH-meter (Hanna Instruments, Eibar, Spain) with a combined glass electrode (FiveEasy, Mettler-Toledo). After each measurement the electrode was calibrated using a pH 7.0 buffer (Merck, Germany).

### 2.4. Water Activity Measurement 

A HygroPalm 23-AW-A water activity analyzer (ROTRONIC AG, Bassersdorf, Switzerland) with a system of automatic registration of water evacuation from an individual sample was used for measurements. Before each measurement, the chamber was dried to a water activity of 0.06. Water activity was measured at room temperature.

### 2.5. Texture Profile Analysis (TPA) 

The texture profile was analyzed with a TA-XT2i texture analyzer (Stable Micro Systems, LTD, Surrey, UK). Cylinders 20 mm × 13 mm (height × diameter) in size were cut from the sausages and compressed. A double compression cycle test was carried out up to a 50% strain of the original height, using a plastic plunger with a diameter of 50 mm and a 0.1 s time interval between the two compression cycles. Force–time deformation curves were obtained with a 25 kg load cell applied at a cross-head speed of 5.0 mm s^−1^. The other operating conditions of the apparatus were as follows: pre-test speed—2.0 mm s^−1^; post-test speed—5.0 mm s^−1^; data acquisition rate—200 PPS, and applied force—10 g. The texture parameters were expressed as hardness (N), cohesiveness, springiness (mm), gumminess (N) and chewiness (N × mm). All the samples were replicated 10 times at room temperature.

### 2.6. Microbiological Analysis

The sausage samples were analyzed microbiologically. The total count of microorganisms was measured according to the PN-EN ISO 4833-1:2013-12 [26] standard and the count of mesophilic lactic acid bacteria was measured according to the PN-ISO 15214:2002 [27] standard.

### 2.7. Sensory Assessment

The finished products were assessed by a nine-member trained panel (7 women aged 30–40 and 2 men aged 36 and 40). They had been trained in the ‘Sensory Analysis of Meat and Meat Products. Selecting the Sensory Team Leader’. Sample preparation for sensory analysis: Cold assessment: the sausages were cut into equal pieces of 3 cm in length; the longitudinally cut sausage was used for the assessment of the cross section.Hot assessment: the sausages were immersed in boiling water for 3 min, then they were cut into pieces of 3 cm in length.

Each participant received 3 pieces of the product for cold and hot evaluation and a longitudinally cut sausage for evaluation of the cross-section. The consumption of a heated product is preferable, but it is also served cold very often.

The intensity of selected characteristics of the products was assessed quantitatively according to a predetermined scale based on the scaling method described in the PN-ISO 4121:1998 [28] standard. The method uses a predetermined scale to quantify the intensity of a selected trait of the product. The sensory panel evaluated three coded samples (the control sample and the samples with the fish oil additives) one by one in random order. The results of the evaluation were analyzed to determine differences between individual samples. The following sensory characteristics of the finished products were assessed: external appearance, external color, cross-sectional appearance, cross-sectional color, smell of the cold product, taste of the cold product, consistency of the cold product, smell of the heated product, taste of the heated product, consistency of the heated product. The individual sensory characteristics were rated according to a five-point scale. The following degrees of intensity of the tested traits were assigned: very attractive, distinctive, pattern (5 points), attractive, with small deviations (4 points), average, with noticeable deviations (3 points), inadequate, with significant deviations (2 points), unacceptable (1 point). In order to better present the overall evaluation and differences between the samples the sensory acceptance percentage was calculated. The mean score awarded by the trained panel was calculated as the percentage of the total score for the ten discriminants assessed in a sample in relation to the maximum possible score, i.e., 50 points, which was 100%. The products underwent a sensory assessment on days 1 and 21 of refrigerated storage.

### 2.8. Statistical Analyses

The Statistica 13.1 software (StatSoft, Tulsa, OK, USA) was used for statistical tests. Differences were considered significant at *p* < 0.05. All the tests were replicated at least three times. Multivariate analysis of variance (ANOVA) was used to assess the influence of the type of additive, storage time and packing method on the water activity and pH. The Kruskal–Wallis test was applied to investigate the influence of independent variables on the growth of the microorganisms under analysis and the results of the sensory analysis. Principal component analysis (PCA) was applied as the first step of data analysis to visualize information and detect patterns in the data.

## 3. Results and Discussion

### 3.1. Basic Composition

The basic chemical composition of the model sausages is shown in Table 1. The fish additive did not significantly affect the basic composition of any of the enriched products. These results were expected, because only small amounts of fish oil and microcapsules were added to the poultry stuffing, i.e., 7.1 g kg^−1^ and 11.9 g kg^−1^, respectively. The results of our study were similar to those of an earlier study on the same type of sausages [5]. There were similar results of the study on chicken nuggets conducted by Jiménez-Martín et al. [29].

Although there were no statistically significant differences (*p* < 0.05), the water content in the samples with liquid and microencapsulated oil decreased slightly. In other studies, the water content in pork salchichón and fresh and cooked pork burgers enriched with fish oil microcapsules [30,31] was also lower than in the control samples. The authors indicated that the effect was caused by the addition of extra dry matter. At the same time, the protein concentration in the product with the microcapsules (18.3%) was slightly higher than in the control sample (17.8%). A similar result was observed by Solomando et al. [32] in their study on cooked and dry-cured sausages. The result may have been influenced by the collagen shell in which the fish oil was contained. Josquin et al. [33] observed that the protein content seemed to be significantly influenced by the encapsulated fish oil additive, which was probably due to the extra dry matter contained in it.

### 3.2. pH Analysis

The initial pH values were similar in all the variants of sausages and ranged from 5.83 to 5.96 (Figure 1). They did not differ significantly (*p* > 0.05) from the results of the earlier study conducted on the same samples of vacuum-packed sausages, where the average values measured on the first day ranged from 5.82 to 5.95 [5]. In this study, there was a slightly higher pH in the samples with fish oil. Other authors observed a similar tendency in their studies [4,7]. In a present study, this increase may have been caused by the protein component and the lipid material of the oil-in-water emulsion. However, Muguerza et al. [34] did not observe such changes. The sausage with the microencapsulated oil additive was characterized by the lowest average pH, i.e., 5.84, which was close to the pH of the control sample. Pelser et al. [35] found no differences in the pH value between the control sample of fermented sausage and the sample modified with oil microcapsules (*p* < 0.05).

Throughout the entire storage period there were no significant changes (*p* > 0.05) in the pH of the control VP sample. The value of this parameter increased slightly in the samples with the liquid oil additive on the 21st day of storage. The pH in the vacuum-packed sample was 6.15, whereas in the MAP sample it was 5.99. The control samples of sausages and the ones with the microencapsulated oil additive packed in the modified atmosphere were characterized by the lowest average pH values (5.80) in the last period of storage, as compared with the other variants of sausages.

The mixture of gases used for modified atmosphere packing changes its physicochemical properties during the storage of meat or sausages. One of the gases is CO_2_, which dissolves during storage. In consequence, the pH value may decrease. During the 21-day storage of the model sausages, the CO_2_ content decreased from 27.9% to 20.3%. The decrease in the pH of poultry lunch meat may also be caused by increased production of lactic acid resulting from the metabolism of lactic acid bacteria in the product [36].

### 3.3. Water Activity

The observation of the sausages during their storage showed a leakage inside the package (unpublished data). Its amount increased along with the storage period, regardless of the type of packing. However, the losses resulting from the leakage in the vacuum-packed products were significantly higher (5.81% ± 0.63 on average) than in the modified-atmosphere-packed sausages (1.32% ± 0.22 on average) (*p* < 0.05). This finding is consistent with the results from study of Stasiewicz et al. [37] who observed that after 15 days of the storage of vacuum-packed sausages and sausages packed in a modified atmosphere composed of 80% N_2_ and 20% CO_2_, the losses amounted to 6.8% and 1.52%, respectively. The greater leakage from vacuum-packed sausages results from the effect of negative pressure on the products [38]. The loss of juice from lunch meat may affect the measurement of water activity and the instrumental analysis of the texture and sensory analysis of the model products.

Figure 2 shows the values of the equilibrium water activity (A_w_) in the model sausages. The average initial water activity values (on the 1st day) in the sausages (CO, FO, MC) were similar and amounted to 0.859 ± 0.001. Neither the form of the fish oil additive nor the packing method had a significant influence on the A_w_ values (*p* < 0.05). 

According to Fernandez-Fernandez et al. [39], the packing method also did not have a significant effect on the A_w_ values in packaged chorizo, nor did it have a significant influence on water activity in salchichón packed under different gas conditions [14]. In the subsequent storage periods, water activity decreased gradually in all the samples. After 21 days of storage of the control samples and the samples with the liquid oil additive, the trend slowed down. There was a further slight decrease in the A_w_ value in the sausages with the microencapsulated fish oil additive, but it was not statistically significant (*p* > 0.05). In general, as far as the sausages with the liquid oil additive are concerned, A_w_ was the lowest in comparison with the control samples (CO) and the ones with microcapsules (MC). This may have been caused by the fact that the hydrophobic properties of the oil added to the stuffing caused a slight displacement of water from the tested system. The presence of a hydrogel shell from microcapsules may have caused the absorption of a small amount of water from the system and increased water activity in these sausage samples. At the same time, a slightly higher protein content in these samples increased the competitiveness of protein–protein interactions in relation to protein–water interactions. On the one hand, a more compact spatial structure of the experimental systems was formed as a result of heat treatment. On the other, a small proportion of the water which was not involved in the structuring of the hydrocolloid-fat phase may have remained outside the protein matrix in the form of free water, which caused the A_w_ value to increase [5]. Therefore, it is possible to suggest that the differences in the water activity level were caused by the physical state of water resulting from the molecular structure of the system under study.

The PCA method allows us to present the relationships between the samples in one graph. Figure 3 shows the results of principal component analysis (PCA) based on correlations applied to the samples stored for three weeks. PC 1 (36.39%) and PC 2 (21.93%) accounted for 58.32% of the total variance. There was no correlation between pH and water activity, nor between water activity, the type of sample and the packing method. However, there was a negative correlation between pH and the type of sample and the packing method. Moreover, the analysis showed that pH was not correlated with the storage time, while water activity tended to decrease along with the storage time. The presented figure shows the relationship between the dependent variables (pH, Aw) and independent variables (type of sample, storage time, packing). Points representing the type of sample and packing are placed relatively close to each other which may indicate a positive correlation. Unfortunately, these are independent variables, and therefore cannot describe the correlation.

### 3.4. Texture Analysis

Meat batter is a complex system. The texture of finely comminuted lunch meats is determined by the reaction of the spatial protein matrix to mechanical interactions and the physical condition of the meat emulsion, which is the continuous phase of the system. The share and structural parameters of the dispersed phase consisting of fragments of the muscle tissue are less likely to be responsible for the texture [40]. The replacement of animal fat with fat of a different origin may involve several changes in the textural (e.g., hardness, chewiness, etc.) and rheological properties of the product (e.g., modulus of elasticity). The characteristics of the texture of meat products may be influenced by the choice of fat source. These effects can also be attributed to the degree of saturation of fatty acids and the size and distribution of globules of fat in the system under study [41,42].

The instrumental texture analysis (TPA) of the experimental sausages that was conducted during the study enabled five mechanical discriminants to be determined: hardness, springiness, cohesiveness, gumminess, and chewiness. There was an increase in the compressive force, i.e., hardness, in the sausages containing microencapsulated fat (MC) compared with the other two sausage variants (Figure 4). This tendency was observed in all the periods of cold storage for the experimental sausages and in both packing variants. An increase in the hardness of poultry sausages samples with the microencapsulated oil additive was also observed in an earlier study by Stangierski et al. [5]. This fact can be explained by collagen from the shell of fat-containing microcapsules contributing to the formation of a spatial protein matrix maintaining the water–fat emulsion [43]. The hardness of all the model sausages increased along with the storage time regardless of the packing method. It may have been caused by the increasing leakage from the sausages during the storage. The effect may also have been caused by an increase in the density of the spatial protein matrix due to the respiralization of polypeptide chains during refrigerated storage [44]. There were different results in the study by Pavlik et al. [45], who investigated the effect of microencapsulated n-3 fatty acid on the qualitative properties of dry sausages. On the 21st day of storage, the sausages’ hardness was not affected by the additive of microencapsulated oil. However, it is important to note that a different type of sausage was tested.

The springiness of all the poultry sausages was similar, i.e., 0.91 ± 0.01 on average, throughout the entire storage period and in both packing variants (*p* > 0.05). The equal value of this determinant of the texture of poultry sausages can be justified by similar fat content [41]. Springiness is a measure of the ability of a system to regain its original shape after the force deforming the system has been removed. The elastic reaction is caused by the protein network formed as a result of heating, whereas the water–fat emulsion is responsible for the plastic properties, i.e., the dissipation of mechanical energy [46]. The lack of significant differences in the springiness of the sausage samples tested in the study may also have been caused by the fact that its value is a component of elastic and plastic traits, which compensate each other.

There were also slight differences in the next texture discriminant, i.e., cohesiveness (*p* > 0.05). The mean value of this discriminant for all the samples was 0.62 ± 0.02. This trait is a measure of the strength of internal bonds in the structure of the system. In general, there were slightly higher values of this texture discriminant in the sausage with the fish oil additive (FO) in the first two periods of refrigerated storage, especially in the vacuum-packed samples. This effect may be explained by the fact that liquid fats disperse more easily and are emulsified more quickly than solid fats. Also, a higher amount of fat in the composition may result in the formation of a more compact continuous phase of the emulsion in the stuffing, which binds all the ingredients of the finished product more strongly when it is heated [5,7]. Cohesiveness did not exhibit an obvious trend in fish sausages. This may have been caused by the fact that other factors such as moisture and water activity are responsible for cohesiveness [31]. In an earlier study conducted on analogous samples of sausages, the samples with the fish oil additive (FO) were characterized by greater cohesiveness than the other experimental systems [5]. The differences in the results may have been caused by differing quality of the raw poultry material, especially the protein and fat content, or differing pH. These parameters may affect the functional properties of meat, such as water absorption, as well as the emulsifying and gelling capacity of proteins

Other texture parameters analyzed in the study were gumminess and chewiness (Figure 5 and Figure 6). The experimental sausage systems were characterized by slight, but statistically significant differences in individual storage periods, which also depended on the packing method (*p* < 0.05). The fish oil additive, especially the oil microcapsules, caused an increase in the gumminess and chewiness of the samples. The greatest increase was observed on the 21st day of storage for the VP sample with the microencapsulated fish oil additive. In general, the highest values of gumminess and chewiness were noted in the samples with the microencapsulated fish oil additive (ME), which were followed by the samples with the liquid oil additive (FO) and the control samples (CO). The MAP samples were characterized by more even gumminess and chewiness than the VP samples in all the storage periods. The results of this study are similar to the findings of the earlier study by Stangierski et al. [5]. The differences in the assessment of the texture parameters may have been caused by changes in the molecular structure of the sausage samples which occurred during their storage, e.g., the degree of water binding in protein systems. It is also necessary to remember that these texture parameters were not determined during measurements, but were calculated on the basis of the values of the original parameters. This fact also resulted in similar courses of changes in these traits, shown in Figure 5 and Figure 6.

In summary, in comparison with the control system, the use of small amounts of the liquid oil or microencapsulated oil additives in the formulation of model stuffing caused slight changes in the abovementioned textural characteristics.

### 3.5. Microbiological Quality

The counts of aerobic bacteria and mesophilic lactic acid bacteria were measured in the samples of poultry sausages. This is a standard procedure to determine the shelf life of meat products during refrigerated storage [36]. Changes in the counts of bacteria during the refrigerated storage of VP and MAP poultry sausages with and without the fish oil additive are shown in Table 2. In the initial storage period (1st day), the count of aerobic bacteria in the samples of modified-atmosphere-packed and vacuum-packed sausages without the fish oil additive amounted to 2.18 ± 0.01 log CFU g^−1^ and 2.81 ± 0.02 log CFU g^−1^, respectively. There were more aerobic bacteria in the vacuum-packed sausages, regardless of the type of sample. The count of aerobic bacteria in all the samples of sausages packed in the modified atmosphere was comparable, regardless of the form of the oil additive used. The difference in the count of bacteria between the modified-atmosphere-packed and vacuum-packed samples of sausages without the additive, the ones with the liquid fish oil additive and the ones with the microencapsulated fish oil additive amounted to 0.63 log CFU g^−1^, 0.53 log CFU g^−1^, and 0.47 log CFU g^−1^, respectively. The most aerobic bacteria were found in the samples of poultry sausages without the fish oil additive, regardless of the packing method, whereas the least bacteria were found in the samples with the microencapsulated fish oil additive. The difference in the count of aerobic bacteria between the samples of sausages without the fish oil additive and the ones with the fish oil microcapsules amounted to 0.14 log CFU g^−1^ for the MAP samples and 0.30 log CFU g^−1^ for the VP samples.

During storage, the count of aerobic bacteria increased both in the samples of sausages packed in the modified atmosphere and in the ones packed in a vacuum (Table 2). The storage time and method of packing the samples had a statistically significant influence on the increase in the count of aerobic bacteria (*p* < 0.05). The count of bacteria in the modified-atmosphere-packed and in the vacuum-packed samples of sausages without the fish oil additive amounted to 4.23 ± 0.01 log CFU g^−1^ and 4.90 ± 0.01 log CFU g^−1^, respectively. On the 21st day of storage, there were more aerobic bacteria in the VP sausage samples, regardless of the form of the fish oil additive used. The test result points to the inhibitory effect of carbon dioxide on aerobic bacteria. The same effect was also demonstrated in a study assessing the influence of modified atmosphere packing on the quality and shelf-life of frankfurter sausages [47]. Martinez et al. [48] found the inhibitory effect of CO_2_-rich atmospheres (60%) on microbial growth in fresh pork sausages. A gas atmosphere containing only carbon dioxide or carbon dioxide combined with nitrogen inhibits the growth of aerobic bacteria such as *Pseudomonas* spp. [49]. On the 21st day of storage, the most aerobic bacteria were found both in the MAP and VP samples of control sausages without the oil additive. The least bacteria were found in the samples with the microencapsulated fish oil additive. The count of bacteria in the VP samples of poultry sausages with the microencapsulated fish oil additive was comparable with the count of bacteria in the MAP samples without the oil additive. The statistical analysis showed that the packing of sausages in the modified atmosphere slowed down the proliferation of aerobic bacteria during storage. Raesi et al. [50] observed that encapsulated fish oil and garlic essential oil added to chicken nuggets significantly delayed not only the oxidation of lipids, but also the microbiological deterioration of samples during storage. The addition of capsules with fish oil and garlic essential oil proved to be more effective than the addition of bulk fish oil.

Immediately after packing (1st day), the count of mesophilic lactic acid bacteria was less than 10 CFU g^−1^, regardless of the type of sample (Table 2). The storage time had a statistically significant effect on the growth of lactic acid bacteria in the sausages. On the last day of storage (21st day) the count of mesophilic lactic acid bacteria increased both in the MAP and VP sausages. The increase ranged from 2.45 log CFU g^−1^ to 3.48 log CFU g^−1^, depending on the type of sample and the packing method. The LAB count in the samples of sausages packed in the modified atmosphere ranged from 3.45 to 3.72 log CFU g^−1^, whereas in the vacuum-packed samples it ranged from 4.08 to 4.48 log CFU g^−1^, depending on the type of sample. Luong et al. [51] found that during storage the LAB count in samples of turkey sausages packed in an atmosphere with 50% CO_2_ and 50% N_2_ was greater than in samples packed in an atmosphere with 70% O_2_ and 30% CO_2_. On the 21st day of storage of the poultry sausages, there were more lactic acid bacteria in the VP samples. This result is consistent with the findings of other authors, and confirms the fact that the growth of lactic acid bacteria in modified atmosphere packages is inhibited by carbon dioxide, especially when the gas concentration exceeds 30% [47]. The lowest count of lactic acid bacteria was found in the samples of sausages containing fish oil microcapsules, regardless of the packing method. The highest count of lactic acid bacteria was found in the samples of sausages with the liquid fish oil additive. The count of lactic acid bacteria in the samples of modified-atmosphere-packed sausages containing fish oil microcapsules was 0.63 log CFU g^−1^ lower than in the vacuum-packed samples containing the same additive. Lactic acid bacteria may produce acids, alcohols, sulfur compounds and aldehydes. Therefore, if these bacteria develop during the storage of products, they may cause unfavorable organoleptic changes, such as a sour taste and smell of the product as well as slime on its surface.

On the last day of storage, the count of bacteria in none of the sausage samples exceeded the recommended safety limit of 7 log CFU g^−1^ [52], which indicates the acceptable microbiological quality of the product. The experiment showed that the modified atmosphere and the microencapsulated fish oil additive resulted in the lowest microbiological contamination of the sausages. The atmosphere with carbon dioxide was more effective than vacuum in reducing the count of bacteria in the poultry sausages with the fish oil additive during their storage. In the final period of storage, lactic acid bacteria were the dominant type of bacteria in both the vacuum-packed and modified-atmosphere-packed sausages. The reduction of the oxygen content to 0.33% in the modified atmosphere package after storage or the removal of oxygen from the vacuum package creates favorable conditions for the growth of lactic acid bacteria. The dominant share of lactic acid bacteria was also observed during the storage of fresh turkey and pork sausages packed in a modified atmosphere [51].

Two principal components were identified from the correlation matrix. They explained 79.55% of total variability (PC1—59.23% of variability, PC2—20.32% of variability). The correlations with input variables are shown in Figure 7. The storage time of the samples was positively correlated with the growth of bacteria. However, the total count of bacteria after 21 days of storage did not exceed the recommended safety limit in any of the samples.

### 3.6. Sensory Assessment

The model poultry sausages underwent sensory analyses when they were cold and then after being heated. These types of lunch meat are consumed both hot and cold. The results of the hedonic test are shown in Figure 8a–d. Neither the enrichment of the meat stuffings with the liquid and microencapsulated fish oil additive, nor the packing method caused significant differences in the batches of sausages under analysis on the 1st day of storage. However, the analysis made by the trained panel showed that the VP and MAP samples with the microencapsulated fish oil additive were rated slightly higher (84.5% on average) than the control sample (83.4% on average) and the samples with the liquid fish oil additive (82.6% on average). As shown in Figure 8a–d, most of the sensory characteristics were very similar to each other, especially during the first day of storage. The biggest, though statically insignificant changes in vacuum packed product were noticed in terms of cross-sectional color, regarding the samples with fish oil and the addition of microcapsules (*p* > 0.05). The difference reached 0.3 points. There were no differences in cross-sectional color when the MAP packaging method was used. In this case, however, there was disparity in texture after heating. Samples with microcapsules scored better than those with fish oil. Similar to previous examples, the difference equaled 0.3 points. For all other sensory characteristics between samples, the differences were not present or on level of 0.2–0.3 points. The total score for all graded characteristics for the VP samples of the control sausage was 48.5, for the ones with the liquid fish oil additive—47.9, and for the ones with the oil microcapsules—48.9. The overall scores for the MAP samples were: 48.3, 48.0, and 48.9, respectively. 

Storage of the model sausages for 21 days caused a slight decrease in most of the sensory parameters. On average, the overall score decreased by 0.8 points for the vacuum-packed (VP) samples and by 1.0 point for the modified-atmosphere-packed samples (MAP). The lower values of the sensory discriminants were mainly related to smell and taste. However, these changes were statistically insignificant (*p* > 0.05). Importantly, the score of the panel did not indicate any signs of unwanted fish aftertaste in the samples with the fish oil. There were also minimal changes in their texture. This may have been caused by leaks during storage. 

Evaluation panel scores indicated some deterioration of VP sausages appearance resulting from their slight deformation. Unfortunately, VP may cause this defect. Based on the sensory evaluation it is difficult to clearly indicate which sausage variant and packing method were the most beneficial. 

Figure 9 shows that the principal component analysis was built from all instrumentally obtained texture traits and selected discriminants of the sensory analysis (cold consistency and consistency after heating). PC 1 (35.48%) and PC 2 (16.04%) explained 51.52% of the total variance. There was a positive correlation between springiness, cohesiveness and the type of sample. Chewiness and gumminess formed separate clusters, thus indicating a positive correlation.

## 4. Conclusions

Results obtained in this study indicated that both refined liquid and microencapsulated fish oil added to the model poultry sausages did not deteriorate their functional attributes. Since the aim of the research was to obtain poultry frankfurter sausages with favorable qualitative characteristics and enriched with fatty acids from the n-3 family, this finding seems to be the most important. On the first day of storage, sensory analysis did not show any significant differences between the samples. After 21-days storage of the poultry sausages there was a slight decrease in some of the sensory parameters, e.g., color, smell, taste. 

The microencapsulated fish oil additive slightly increased the strength-related properties of the resulting sausages, which was confirmed by the instrumental analysis. This change was favorably assessed by the evaluation panel. 

The experimental VP samples had a higher content of aerobic bacteria and lactic acid bacteria. The carbon dioxide applied for sausage packaging inhibited the growth of bacteria. It is important to note that after 21 days of storage, the recommended safety limit of bacteria was not exceeded in any of examined samples of sausages. This observation confirms good microbiological quality of the finished products.

On the assumption of favorable economic analysis, it seems reasonable to conclude finally, that meat processors should consider liquid rather than microencapsulated oil additives by the successful elaboration of poultry sausages enriched with fatty acids from the n-3 family.

## Figures and Tables

**Figure 1 sensors-21-02653-f001:**
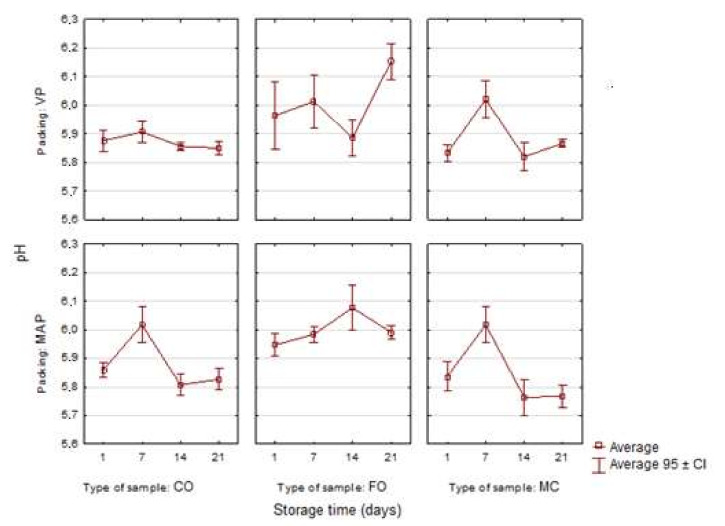
Changes in the pH value in the poultry sausages depending on the type of sample, packing and storage time (*n* = 3).

**Figure 2 sensors-21-02653-f002:**
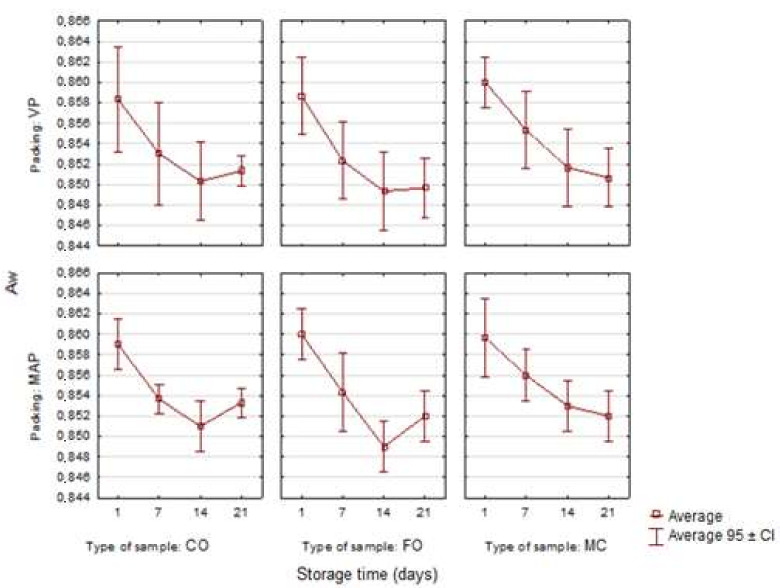
Changes in the water activity (A_w_) in the poultry sausages depending on the type of sample, packing and storage time (*n* = 3).

**Figure 3 sensors-21-02653-f003:**
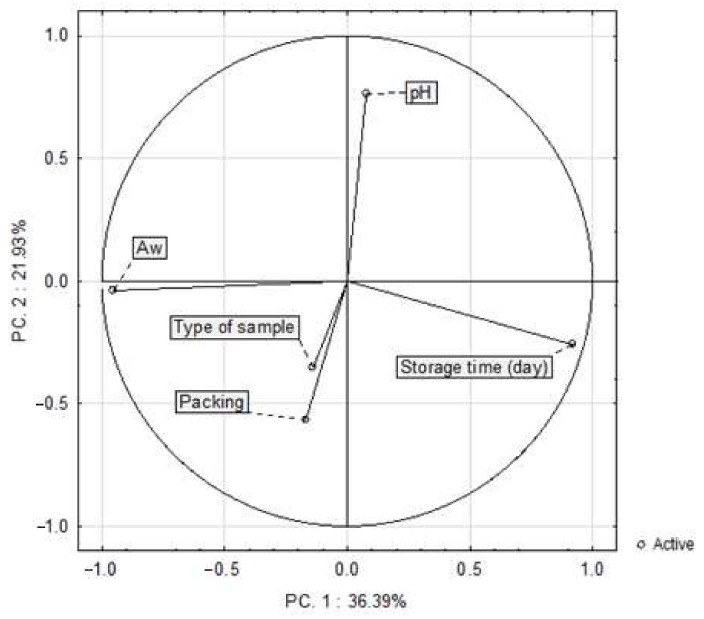
The principal component and load analysis in the analysis of water activity (A_w_) and pH depending on the type of sample, packing and storage time (*n* = 146).

**Figure 4 sensors-21-02653-f004:**
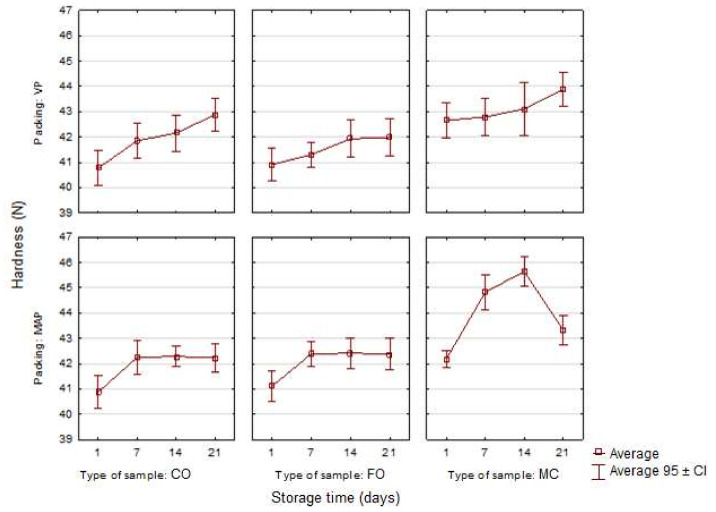
Changes in the hardness in the poultry sausages depending on the type of sample, packing and storage time (*n* = 10).

**Figure 5 sensors-21-02653-f005:**
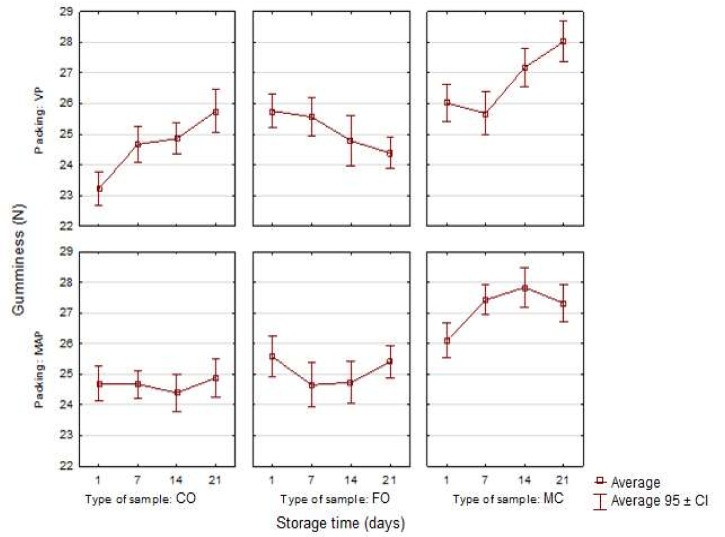
Changes in the gumminess in the poultry sausages depending on the type of sample, packing and storage time (*n* = 10).

**Figure 6 sensors-21-02653-f006:**
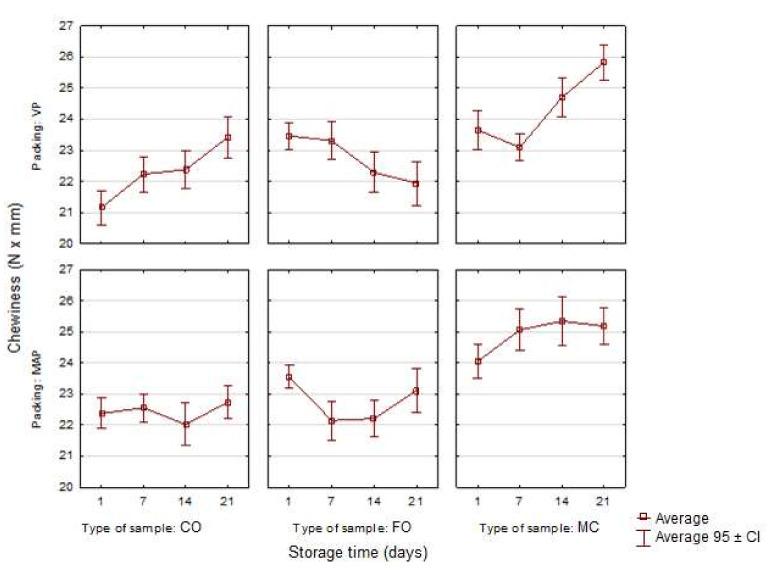
Changes in the chewiness in the poultry sausages depending on the type of sample, packing and storage time (*n* = 10).

**Figure 7 sensors-21-02653-f007:**
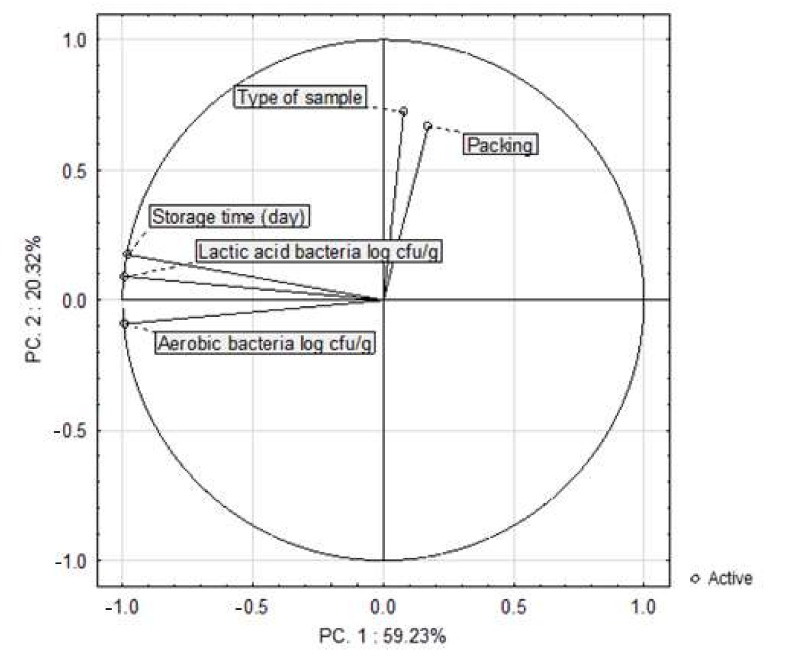
Changes in the value of the total colony count (TCC) and the count of mesophilic lactic acid bacteria (LAB) in the poultry sausages depending on the type of packing and time of storage (*n* = 144).

**Figure 8 sensors-21-02653-f008:**
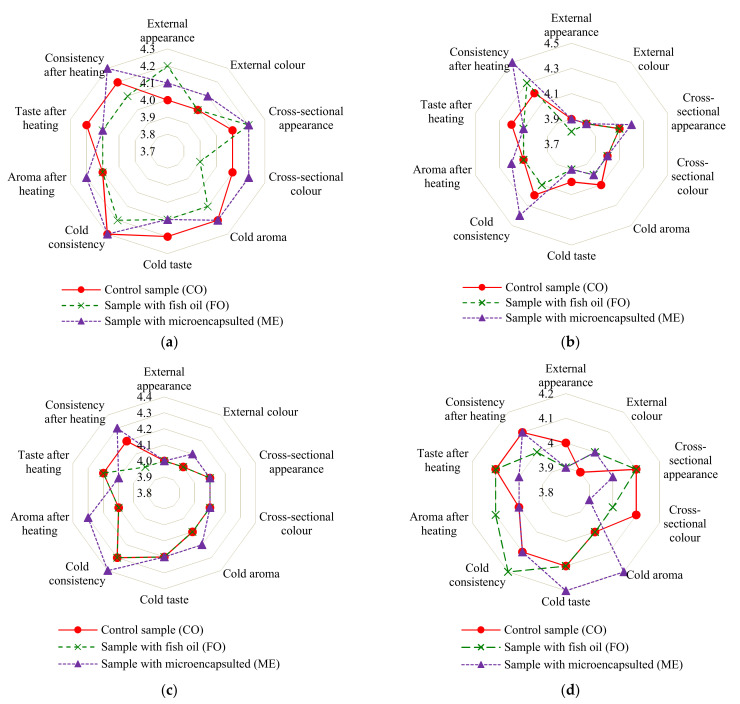
The scores of the trained panel for the intensity of the sensory characteristics of the model systems of chicken sausages: vacuum packed (**a**) the 1st and (**b**) 21st day of storage; modified atmosphere packed (**c**) the 1st and (**d**) 21st day of storage (*n* = 9).

**Figure 9 sensors-21-02653-f009:**
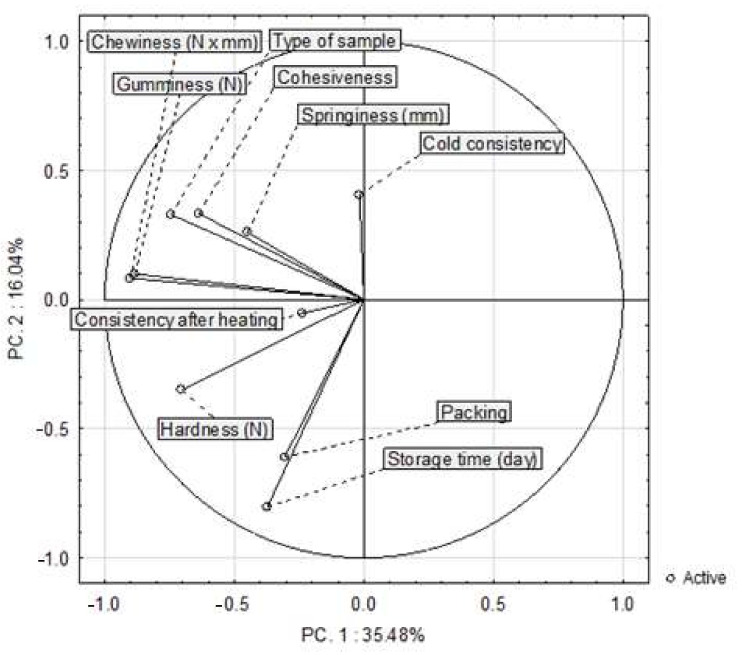
Changes in the value of the sensory and textural analysis in the poultry sausages depending on the type of sample, packing and time of storage (*n* = 960).

**Table 1 sensors-21-02653-t001:** The basic chemical composition of the poultry sausages.

Chemical		Type of Sample	
Composition (%)	Control (CO)	Fish Oil (FO)	Microencapsulated (MC)
Moisture	67.4 ± 0.7 ^a^	66.6 ± 0.7 ^a^	66.2 ± 0.6 ^a^
Protein	17.8 ± 0.3 ^a^	17.6 ± 0.3 ^a^	18.3 ± 0.4 ^a^
Fat	10.2± 0.5 ^a^	11.5 ± 0.6 ^a^	11.0 ± 0.6 ^a^
Ash	1.9 ± 0.1 ^a^	2.0 ± 0.1 ^a^	1.9 ± 0.1 ^a^

^a^—No statistically significant differences between means in the same rows (*p* < 0.05; mean ± standard deviation; *n* = 3).

**Table 2 sensors-21-02653-t002:** Changes in the value of the total colony count (TCC) and the count of mesophilic lactic acid bacteria (LAB) in the poultry sausages depending on the type of packing and storage time.

Storage Time	Microbiological		VP			MAP	
(Days)	Analysis	CO	FO	MC	CO	FO	MC
1	TCC (log_10_ CFU g^−1^)	2.18 ± 0.01 ^aA^	2.11 ± 0.02 ^aA^	2.04 ± 0.02 ^aA^	2.81 ± 0.00 ^aB^	2.64 ± 0.00 ^aB^	2.51 ± 0.01 ^aB^
	Mesophilic LAB (log_10_ CFU g^−1^)	1.00 ± 0.00 ^aA^	1.00 ± 0.00 ^aA^	1.00 ± 0.00 ^aA^	1.00 ± 0.00 ^aA^	1.00 ± 0.00 *^a^*^A^	1.00 ± 0.00 ^aA^
21	TCC (log_10_ CFU g^−1^)	4.23 ± 0.01 ^bC^	4.18 ± 0.00 ^bC^	3.94 ± 0.00 ^bC^	4.90 ± 0.01 ^bD^	4.36 ± 0.02 ^bD^	4.23 ± 0.00 ^bD^
	Mesophilic LAB (log_10_ CFU g^−1^)	3.57 ± 0.01 ^bB^	3.72 ± 0.01 ^bB^	3.45 ± 0.02 ^bB^	4.15 ± 0.03 ^bB^	4.48 ± 0.01 ^bB^	4.08 ± 0.04 ^bB^

^a,b^—Means values in columns denoted by different letters differ statistically significant (within one group of bacteria) (*p* < 0.05; mean ± standard deviation; *n* = 3). ^A–D^—Means values in rows denoted by different letters differ statistically significant (within one group of bacteria) (*p* < 0.05; mean ± standard deviation; *n* = 3).

## Data Availability

Not applicable.

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
