# Peer review of "The Influence of Packing Methods and Storage Time of Poultry Sausages with Liquid and Microencapsulated Fish Oil Additives on Their Physicochemical, Microbial and Sensory Properties"

_sensors, 2021, doi:10.3390/s21082653_

Round 1
Reviewer 1 Report
Meat products are an excellent matrix for incorporating other ingredients. In general I think this research if of interest.
1. I would suggest that in the introduction/discussion you could including the following articles.
Olmedilla-Alonso, B., Jiménez-Colmenero, F., & Sánchez-Muniz, F. J. (2013). Development and assessment of healthy properties of meat and meat products designed as functional foods. Meat science, 95(4), 919-930.
Domínguez-Soberanes, J., Escalona-Buendía, H. B., Flores-Nájera, A., González, A. D. L. R., Chávez-Vela, N., & Rodríguez-Serrano, G. (2017). Product design of a ready to eat Sous vide marinated pork meat based on consumer acceptability and prebiotic index. Revista Mexicana de Ingeniería Química, 16(2), 491-501.
Teixeira, A., & Rodrigues, S. (2020). Consumer perceptions towards healthier meat products. Current Opinion in Food Science.
Shan, L. C., De Brún, A., Henchion, M., Li, C., Murrin, C., Wall, P. G., & Monahan, F. J. (2017). Consumer evaluations of processed meat products reformulated to be healthier–A conjoint analysis study. Meat science, 131, 82-89.
Schnettler, B., Ares, G., Sepúlveda, N., Bravo, S., Villalobos, B., Hueche, C., & Adasme-Berríos, C. (2019). How do consumers perceive reformulated foods after the implementation of nutritional warnings? Case study with frankfurters in Chile. Food quality and preference, 74, 179-188.
2. In the methodology I would suggest you to indicate how many samples were analyzed in water activity, pH, microbiological analysis, etc.
3. Sensory analysis is not very clear and many questions arrise. Probably you could explain in a more detailed way the profile of the participants, the sample preparation and the measurements of sensory attributes and liking scores.
Were the measurements made by a trained panel or consumers? How many? What age? Were there international guidelines taken into consideration for the sensory analysis?
On line 523, how was the average calculated? Is this the overall liking score, how was it calculated?How is the food product prefered, cold or heated?
For this purpose I recommend these articles as references:
Sánchez, C. N., Domínguez-Soberanes, J., Escalona-Buendía, H. B., Graff, M., Gutiérrez, S., & Sánchez, G. (2019). Liking product landscape: going deeper into understanding consumers’ hedonic evaluations. Foods, 8(10), 461.
Samant, S. S., & Seo, H. S. (2019). Using both emotional responses and sensory attribute intensities to predict consumer liking and preference toward vegetable juice products. Food Quality and Preference, 73, 75-85.
Nguyen, Q. C., & Varela, P. (2021). Identifying temporal drivers of liking and satiation based on temporal sensory descriptions and consumer ratings. Food Quality and Preference, 89, 104143.
Ares, G., & Varela, P. (2017). Authors’ reply to commentaries on Ares and Varela. Food Quality and Preference, 61, 100-102.
5. In line 397 I would write formulation instead of recipe.
6. In line 482, this is very good to know!
Reviewer 2 Report
In general, improved nutritional quality of meat products is an interesting area of research that makes this study valuable. However, Manuscript requires detailed revision in order to clarify the aim of the study, thus, improvement of the language is needed. Thus, to understand how this study will solve specific problem, Abstract should be rewritten in order to provide better overview of the importance of present study, as it is quite vague and could be more focused on the effect of fish oil on sausage quality. In addition, sensory section should be improved and written to understand what were sensory traits, positive/negative of the treatment and packaging. In addition, sensory data could be better described to give clear overview what are the results from consumer study and what from the test performed with the trained panel. Conclusion should be improved as well.

Reviewer 3 Report
Although the topic of enriching sausages with various fats, including fish oil, is already known, the experiment described by the authors seems interesting.
The background introduces the topic of poultry sausage technology, and the research methods given are well described. In the opinion of the reviewer, however, a too narrow scope of research was selected.
I think it is a big loss that the authors have carried out storage analyzes without assessing the hydrolytic changes taking place in the sausages. The evaluation of oxidative stability, which can be carried out e.g. by means of the simple TBARS test, was also omitted. The authors investigated the changes taking place during the storage of sausages, which were enriched with oxidatively unstable fat, therefore it would be reasonable to determine whether the product does not become too rancid after reformulation. Or maybe these changes in fat oxidation have a negative impact on organoleptic evaluation of sausages?
I wonder about the validity of the study of the number of lactic acid bacteria. Why did the authors choose this parameter? What does this bring to work? Wouldn't it be better to mark bacteria from the Enterobacteriaceae group, which is an indicator of the hygiene of meat products?
I also believe that the organoleptic assessment of the product may not be adequate. Why did the authors not use the multiple comparison method, where the control sample is the standard, and the samples with vegetable oil additives would be assessed as better or worse than the control?
I would be grateful for dispelling my doubts.
Round 2
Reviewer 2 Report
L249-251: They did not differ numerically significantly from the results of the earlier study conducted on the same samples of vacuum-packed sausages, where the average values measured on the first day ranged from 5.82 to 5.95 [5].
Please specify if they differed or not?
L253-255: “In these researchers’ opinion,”…. In a present study..
L264-265: Throughout the entire storage period there were no statistically significant changes (p<0.05) in the pH of the control VP sample.>>0.05).
L282-284: “much greater”… is this significantly higher or numerically, please clarify?
L285: “These researchers”……. 284-288 lines are pointing on specific research, so please rewrite these sentences.
L307-308: …”but it was not statistically significant (p<0.05) in the pH of the control VP sample.>>0.05)”
L323-330: Please, explain the outcome of PCA, what was correlated positively and negatively correlated.
L371-372: was it significant difference or not? The sentence is not in agreement with given p-value.
L381-382: was it significant difference or not? The sentence is not in agreement with given p-value.
L403: “The next texture characteristics analysed in the study”…. Another texture parameter…
L453 and 455: mean ± standard deviation
L470: please use consistent terms if 21st day of storage use it through the text instead “last day of storage”
L527: “This tendency might seem worrying”…. However…
L542-545: instead of total score, please focus on what attributes in what products were significantly better/worse?
L560: “However, these changes were statistically insignificant (p<0.05).” This is not correct, again p-value is showing that the change was significant and its written opposite!
L569-570: rewrite the sentence…
L580-584: both sentence needs to be rewritten.
L589: remove the sentence as its not result of your study.
L592: “the recommended limit of bacteria”… is this correct way to say?
In general, conclusions should be more clear and better written.
Author Response
Thank you very much for your valuable comments. The manuscript was corrected following all points raised by the Reviewer. We indicated the changes in the text by using green coloured letters.
Kind regards,
Jerzy Stangierski and coauthors
L249-251: They did not differ numerically significantly from the results of the earlier study conducted on the same samples of vacuum-packed sausages, where the average values measured on the first day ranged from 5.82 to 5.95 [5].
Please specify if they differed or not?
The p-value was specified.
L253-255: “In these researchers’ opinion,”…. In a present study..
The sentence has been rewritten.
L264-265: Throughout the entire storage period there were no statistically significant changes (p<0.05) in the pH of the control VP sample.>>0.05).
The sentence has been changed.
L282-284: “much greater”… is this significantly higher or numerically, please clarify?
The sentence has been changed.
L285: “These researchers”……. 284-288 lines are pointing on specific research, so please rewrite these sentences.
The sentence has been rewritten.
L307-308: …”but it was not statistically significant (p<0.05) in the pH of the control VP sample.>>0.05)”
The p-value was corrected.
L323-330: Please, explain the outcome of PCA, what was correlated positively and negatively correlated.
The text has been completed.
L371-372: was it significant difference or not? The sentence is not in agreement with given p-value.
The springiness was not statistically significant (p>0.05). The p-value was corrected.
L381-382: was it significant difference or not? The sentence is not in agreement with given p-value.
The p-value was corrected (p>0.05).
L403: “The next texture characteristics analysed in the study”…. Another texture parameter…
The sentence was rewritten.
L453 and 455: mean ± standard deviation
The sentences have been changed.
L470: please use consistent terms if 21st day of storage use it through the text instead “last day of storage”
The sentence was rewritten.
L527: “This tendency might seem worrying”…. However…
The sentence was rewritten.
L542-545: instead of total score, please focus on what attributes in what products were significantly better/worse?
The text has been rewritten.
L560: “However, these changes were statistically insignificant (p<0.05).” This is not correct, again p-value is showing that the change was significant and its written opposite!
The p-value was corrected.
L569-570: rewrite the sentence…
The sentence has been rewritten.
L580-584: both sentence needs to be rewritten.
The sentence was removed.
L589: remove the sentence as its not result of your study.
The sentence was removed.
L592: “the recommended limit of bacteria”… is this correct way to say?
Indeed "the recommended limit of bacteria"...isn't correctly used;
in this sentence the word "safety" has been omitted.
The correct way is: " the recommended safety limit of bacteria"
In general, conclusions should be more clear and better written.
Conclusions has been rewritten.
Reviewer 3 Report
Based on the authors' responses to the review, I decided to accept the manuscript in the current version.
Author Response
Thank you for accepting our article.
Best regards,
Jerzy Stangierski and coauthors.